# ReaSCAN: Compositional Reasoning in Language Grounding

**Zhengxuan Wu**[*]
Stanford University
wuzhengx@stanford.edu

**Elisa Kreiss**[*]
Stanford University
ekreiss@stanford.edu

**Desmond C. Ong**
National University of Singapore
& IHPC, A*STAR
dco@comp.nus.edu.sg

**Christopher Potts**
Stanford University
cgpotts@stanford.edu

## Abstract

The ability to compositionally map language to referents, relations, and actions is an essential component of language understanding. The recent gSCAN dataset (Ruis et al. 2020, *NeurIPS*) is an inspiring attempt to assess the capacity of models to learn this kind of grounding in scenarios involving navigational instructions. However, we show that gSCAN's highly constrained design means that it does not require compositional interpretation and that many details of its instructions and scenarios are not required for task success. To address these limitations, we propose **ReaSCAN**, a benchmark dataset that builds off gSCAN but requires compositional language interpretation and reasoning about entities and relations. We assess two models on ReaSCAN: a multi-modal baseline and a state-of-the-art graph convolutional neural model. These experiments show that ReaSCAN is substantially harder than gSCAN for both neural architectures. This suggests that ReaSCAN can serve as a valuable benchmark for advancing our understanding of models' compositional generalization and reasoning capabilities.

## 1 Introduction

Natural languages are *compositional* [1, 2, 3] and *grounded* [4, 5, 6]; the meanings of complex phrases are derived from their parts, and meaning itself is defined by a mapping from language to referents, relations, and actions. It is therefore vital that we push NLP systems to be grounded and compositional as well. However, the major benchmarks in the field right now mostly do not support rich grounding, and it is often unclear whether they support learning compositional structures, as evidenced by their common failures at simple adversarial tests involving compositionality [7, 8, 9].

There are several benchmarks for testing compositional generalization [10, 11, 12, 13, 14]. SCAN [12] focuses on compositionality in the area of interpreting navigational instructions. Building off SCAN, Ruis et al. [14] propose a grounded version of SCAN called gSCAN, in which agents have to ground navigation commands in a grid world in order to identify the correct referent. gSCAN supports learning in idealized scenarios involving navigational instructions, and it seeks to probe for compositionality. The design is simple and flexible, making it a potentially valuable benchmark and a source for insights into how to design robust tests of language understanding.

However, we find that gSCAN has three major limitations: (1) its set of instructions is so constrained that preserving the linguistic structure of the command is not required; (2) the distractor objects in its

---

[*]Equal contribution.

35th Conference on Neural Information Processing Systems (NeurIPS 2021) Track on Datasets and Benchmarks.

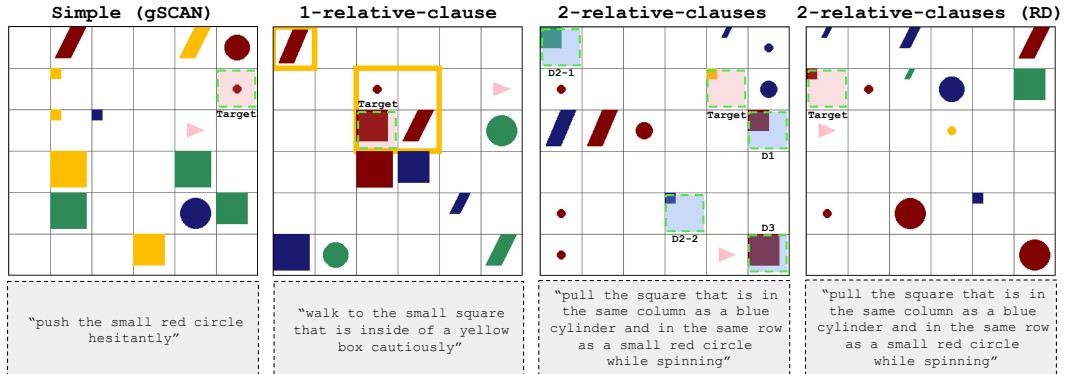

**Figure 1:** Four command–world pairs for different command patterns. ReaSCAN's simple command is equivalent to gSCAN commands [14] but the structure of the sampled grid world differs. RD indicates that distractors are randomly sampled. A sample of potential referent targets is highlighted with green dashes. The actual target for the given the command is shaded in red and its direct distractors are shaded in blue.

grounded scenarios are mostly not relevant for accurate understanding; and (3) in many examples, not all modifiers in the command are required for successful navigation, which further erodes the need for compositional interpretation and inflates model performance scores.

In this paper, we propose **ReaSCAN**, a benchmark dataset that builds off gSCAN and addresses its limitations. Figure 1 provides examples and a comparison with gSCAN. We establish that ReaSCAN requires both compositional language interpretation and complex reasoning about entities and relations. Like gSCAN, ReaSCAN is algorithmically generated, which allows us to vary the difficulty of the learning problems we pose and thus diagnose model limitations with precision. In addition, we introduce a range of complex distractor sampling strategies which, in case of incorrect target identification, can help pinpoint which failure in command understanding led to the error. This allows us to show that challenging distractors can severely impact performance in this task.

We assess two models on ReaSCAN: a multi-modal baseline and a state-of-the-art graph convolutional neural model. These experiments show that ReaSCAN is substantially harder than gSCAN for both neural architectures, and they verify that we can modify the difficulty of learning tasks in the desired ways to achieve fine-grained insights into model performance and model limitations. This suggests that ReaSCAN can serve as a valuable benchmark for advancing our understanding of models' compositional generalization and reasoning capabilities in linguistic tasks. We hope also that the general techniques used to move from gSCAN to ReaSCAN can be applied more generally in the design of future benchmarks for assessing grounded, compositional language use.

## 2 Related Work

There are a variety of efforts underway to more deeply understand how neural models ground linguistic cues with visual inputs, including visual question answering [10, 15, 16, 17, 18], image captioning [19, 20, 21], referring expression resolution [12, 14], navigation [22, 23, 24] and program induction and synthesis [25, 26, 27]. Similar to previous synthetic benchmarks, our work aims to provide a controlled environment that can be used to evaluate a neural model's generalization capabilities according to a variety of specific generalization tasks. We focus on evaluating compositional generalization with grounded referring expression resolution.

A number of recent approaches involve generating synthetic datasets to evaluate compositional generalization of neural models [10, 11, 13, 28, 29, 30, 31, 32]. For instance, [31] proposed CLOSURE, a set of unseen testing splits for the CLEVR dataset [10] which contains synthetically generated natural-looking questions about 3D geometric objects. Our work investigates a similar generalization over grounded linguistic inputs in a visual scene but focuses specifically on a model's capability to resolve linguistic compositionality. We evaluate the generalization capabilities of neural models by testing them against unseen compositions of the language input which require grounding in simulated shape worlds.

Models performing well on gSCAN are a promising first test case for ReaSCAN. Numerous approaches have been proposed to handle at least some of the challenges posed by SCAN and gSCAN datasets including novel data augmentation methods and neural architectures [33, 34, 35, 36, 37, 38, 39, 40]. Successful neural models on gSCAN involve compositional neural networks which increase generalizability [38] and language conditioned graph neural networks for encoding objects [39]. While these techniques solve some of the simpler splits in gSCAN involving generalization of novel object attributes [39], we show that they are still ineffective for similar splits of ReaSCAN in Section 6.2. ReaSCAN, therefore, provides a more challenging benchmark, revealing clear shortcomings of current models' generalization capabilities.

## 3 Background: The Grounded SCAN Benchmark (gSCAN)

The gSCAN benchmark is an extension of the SCAN dataset [12] with a focus on grounding actions in a changeable environment. In gSCAN, a grid world containing an agent and several shapes is paired with a command, such as "walk to the red square cautiously". The goal is to generate an action sequence like ⟨left, right, right, left⟩ that lets the agent execute the command in that particular world to reach the referent target. Adverbs like "cautiously" assign specific modes of movement to the overall sequence. gSCAN enables tests for compositional generalization by presenting the model with unseen verb–adverb combinations ("walk cautiously" vs. "push cautiously"), unseen adjective–noun compositions, unseen color–shape feature co-occurrences on objects, and unseen locations for the target referent.

The guiding ideas behind gSCAN seem powerful and relevant, but we identify four ways in which specific design choices reduce the potential of the dataset to achieve its central goals:

**1. Irrelevance of Word Order** Since gSCAN is meant to be a simple synthetic dataset, all commands consist of a verb, a noun phrase consisting of a noun with a potential color and/or size modifier, and an optional adverb. Given this template, the word order of the input command is irrelevant for determining the correct action sequence. The words "walk to the red square cautiously" can be scrambled and still yield a unique order with only a single potential referent. Consequently, Bag-of-Words accounts are in principle sufficient for encoding the gSCAN commands. As a point of contrast, the commands in the earlier SCAN dataset, such as "walk twice and jump thrice", cannot be scrambled in this way without a task-relevant loss of information, and are therefore much more challenging to solve on the command level.

**2. A Limited Test for Linguistic Compositionality** gSCAN includes a test set in which all commands involve a previously unseen referring expression combination (the novel NP "yellow square"), with the goal of seeing whether models can predict the meaning of the whole from its parts "yellow" and "square", which are seen in training. This is a clear test for compositionality. However, unfortunately, the split creation process didn't inherently require an understanding of "yellow" and "square" to be necessary for a unique identification in a specific world. In the split provided by the authors,[1] both the color and shape feature are only required in 62.7% of all test examples. (Color is sufficient in 25.2% of all test examples, shape in 10.6%, and either of the two in 1.4% of all cases.) gSCAN also includes a test split designed to require feature attribute composition: in training, the referent target is never an object with the color feature *red* and shape feature *square*. At test time, only red squares are targets and are referred to with all valid referring expressions (i.e., "(small|big)? red? square"). As in the previous split, color and shape feature in the command are necessary in just 62.5% of all test examples, making it equally unsuitable for investigating linguistic compositionality.

**3. Biased Distractor Sampling** Distractor sampling in gSCAN relies on random selection of all objects that are not mentioned in the command. In general, if the utterance mentions a blue circle, the algorithm creates all possible objects that aren't blue circles. Then, it selects half of them as distractors. There is one exception: if the utterance contains a size modifier (as in "small blue circle"), there will be a big blue circle as a distractor. Due to the distractor sampling design, simple utterances such as "the circle" will only have one distractor, while more complex utterances will have many more. This makes by-chance accuracy dependent on the informativity and complexity of the linguistic expression.

---

[1] https://github.com/LauraRuis/groundedSCAN

**4. Too Few Effective Distractors**  As shown in the first example for gSCAN of Figure 1, the output action sequence stays the same even if we randomly reorder all objects except the referent target. In fact, as long as they don't introduce reference ambiguity, the size, color, and shape of other objects can be modified without any effect on the output action sequence. As a result, grounding is based on essentially two objects, the two red circles. (See Section 4.2 for details about how distractors affect performance.)

In sum, gSCAN provides novel systematic ways of investigating grounded language understanding but it lacks a way to keep investigating the syntactic compositional questions in the command which motivated SCAN. ReaSCAN introduces a more complex command structure that enforces models to retain some linguistic structure to solve it, and contains compositional splits that ensure the necessity of compositional generalization capabilities for the input command. Due to the more complex command structure, this requires elaborate distractor sampling strategies with the goal to make distractors maximally competitive to promote grounding to multiple objects in the world.

# 4 The Reasoning-based SCAN Benchmark (ReaSCAN)

We now introduce ReaSCAN, which seeks to address the above limitations of gSCAN. Like gSCAN, ReaSCAN is a command-based navigation task that is grounded in a grid world containing an agent, a referent target, and a set of distractors, as shown in Figure 1.

Given a command $\mathbf{C}_i$ paired with a corresponding grid world $\mathcal{W}_{i,j}$, the goal is to generate an action sequence $\mathbf{a}_{i,j}$ which contains the actions that the agent needs to take in order to reach the target referent and operate on it. An oracle model learns a mapping $\mathcal{G}$ that formulates $\mathbf{a}_{i,j} = \mathcal{G}(\mathcal{W}_{i,j}, \mathbf{C}_i)$ for $i \in [1, N]$, where $N$ is the number of commands, and $j \in [1, M]$, where $M$ is the number of worlds generated for each command.

Crucially, ReaSCAN extends gSCAN while ensuring two main desiderata: (1) word-order permutations in the command will lead to ambiguities about the intended referent, requiring a model to resolve linguistic structure, and (2) the identity of the referent depends on reasoning about multiple distractor objects in the world. Consider the `2-relative-clauses` example (third from left) in Figure 1. If we scramble the word order of the command by swapping attributes between the second and the third objects, and change them to "small blue circle" and "red cylinder", the referent target changes (e.g., object D1 in the world); additionally, if the model only understands the first relational clause "the same column as a blue cylinder", it may discover multiple referent targets (e.g., object D2-1 in the world). These modifications ensure that understanding ReaSCAN commands requires resolving the syntactic structure of the command, while largely maintaining the simplicity of SCAN and gSCAN.

In the following sections, we discuss the key components of ReaSCAN. We first introduce the process of generating ReaSCAN commands. Next, we describe how commands are grounded with shape worlds, and specifically the distractor sampling strategies. Finally, we propose test splits which provide systematic tests of a model's generalization abilities.[2] We discuss potential ReaSCAN artifacts in Appendix B.

## 4.1 ReaSCAN Command Generation

ReaSCAN commands are constructed with the following regular expression pattern:

> Pattern := `$VV $OBJ (that is $REL_CLAUSE (and $REL_CLAUSE)*)* $ADV?`

where the recursive structure allows commands to contain multiple relative clauses and conjunctive clauses. If there is no relative clause, the resulting commands are comparable to gSCAN commands (e.g., "walk to the red square cautiously"). From the regular expression, commands are created by sampling terms from each class, where classes are indicated by "$" in the pattern as defined in Table 1. For example, we substitute `$REL $OBJ` for `$REL_CLAUSE`, and we can further recursively sample terms from expression classes.

During this process, we also introduce restrictions to avoid ungrammatical and unnatural commands, enforced by rule-based conditional sampling. This way, commands such as "walk to the square that

---

[2]We release the version of ReaSCAN used in this paper, and our code to generate ReaSCAN at `https://github.com/frankaging/Reason-SCAN`.

| Syntax | Descriptions | Expressions |
|---|---|---|
| $VV | verb | {walk to, push, pull} |
| $ADV | adverb | {while zigzagging, while spinning, cautiously, hesitantly} |
| $SIZE | attribute | {small, big}* |
| $COLOR | attribute | {red, green, blue, yellow} |
| $SHAPE | attribute | {circle, square, cylinder, box, object} |
| $OBJ | objects | (a \| the) $SIZE? $COLOR? $SHAPE |
| $REL | relations | {in the same row as, in the same column as, in the same color as, in the same shape as, in the same size as, inside of} |
| $REL_CLAUSE | clause | $REL $OBJ |

**Table 1:** Definitions of syntax used in ReaSCAN command generation.*the actual size of any shape is chosen from {1,2,3,4} as in gSCAN [14].

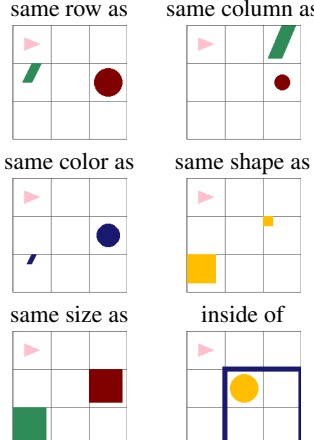

**Figure 2:** Relations.

is in the same color as the red circle" would be excluded, as "walk to the red square" is a shorter and more direct formulation with the same meaning. See Appendix C for details about our rule-based conditional sampling over commands.

In this data creation procedure, both the relative clauses and conjunctive clauses have the flexibility to expand in depth and in width. In this paper, we focus on commands with a maximum of a single conjunction of two relative clauses. In total, we generate the following commands:

- Simple:= $VV $OBJ $ADV? (equivalent to gSCAN commands)
- 1-relative-clause:= $VV $OBJ that is $REL_CLAUSE $ADV?
- 2-relative-clauses:= $VV $OBJ that is $REL_CLAUSE and $REL_CLAUSE $ADV?

We use our framework to generate three separate subsets for each command pattern. We then define random train/dev/test splits for each of the subsets to benchmark difficulty (see Section 6.1 for details), where Simple commands are equivalent to gSCAN commands. As shown in Figure 4, the action sequence length has the same distribution as gSCAN and across all patterns.

## 4.2 ReaSCAN World Generation with Active Distractor Sampling

Similar to gSCAN, we use the open-sourced MiniGym from Open-AI[3] to generate multiple shape worlds for each command. Objects are freely placed in an $n \times n$ grid-world, where we fix $n = 6$. Given a command $\mathbf{C}_i$, objects and their locations are determined as follows: (1) We select objects mentioned in $\mathbf{C}_i$, initialize them with their specified features, and randomly fill underspecified features. For instance, in Figure 3, the command requires the second object to be green and a circle, but its size is not specified and so size is randomly assigned (e.g., here as *big*). (2) The objects are randomly placed on the grid while ensuring the relations expressed in $\mathbf{C}_i$ are true. (3) We sample distractors in a way that ensures that failure to fully understand $\mathbf{C}_i$ has a high likelihood of leading to an incorrect prediction about the target.

As discussed in Section 3, careful distractor sampling is essential for ensuring that our dataset can be used to assess systems for compositionality. Distractors must reliably introduce uncertainty about the identity of the target.

For example, if the target is a small red circle, a large red circle competes with the target in the size dimension, and confusing the distractor with the target would indicate a lack of understanding of the size domain or its composition. Distractors that have little in common with the target are therefore weak distractors. We employ four distractor-sampling methods that ensure a challenging task that can be used to reliably diagnose specific model shortcomings. We exemplify their purpose by referring back to the 2-relative-clauses example (i.e., the third example) in Figure 1.

[3]https://github.com/maximecb/gym-minigrid

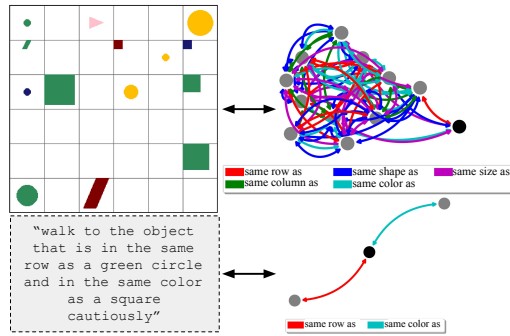

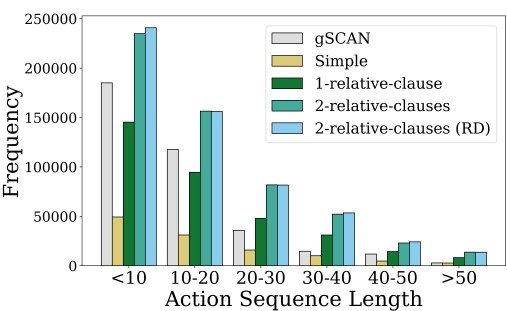

**Figure 3:** Illustration of the conversions between the multi-edge graph and the shape world or the command.

**Figure 4:** Length distributions of action sequences for different datasets.

**Attribute-based distractors** compete with the target if a model struggles with size, color, and shape features. They are created by simulating a change of one of these features in the command and adding objects to the world which make a distractor the plausible target. For instance, if we substitute the shape attribute of the "blue cylinder" to "circle" in the command, the referent target changes (e.g., it could be object D3 in the world). Correctly interpreting the shape attribute becomes crucial for correct target identification.

**Isomorphism-based distractors** become potential targets after word-order permutations of the command. For instance, if we scramble the word order of the command by swapping attributes between the second and the third objects, and change them to "small blue circle" and "red cylinder", the referent target changes (e.g., object D1 in the world). These distractors are crucial to ensure the necessity of linguistic compositionality to solve the task while Bag-of-Words models can maximally achieve chance accuracy.

**Relation-based distractors** ensure that the relative clauses in the command are required to identify the intended target referent. For instance, if the model only understands the first relational clause "the same column as a blue cylinder", other distractors may become the referent target (e.g., object D2-1 in the world); similarly, if the model only understands the second relational clause "the same row as a small red circle", object D2-2 in the world may become the referent target.

For each world, we sample relation-based distractors exhaustively, and we sample at least one attribute-based distractor by randomly selecting one object and perturbing its attribute. For isomorphism-based distractors, we randomly select any pair of objects and swap attributes if applicable. If a distractor-sampling method cannot work for a specific command-world pair, we incorporate **random distractors** by randomly sampling a size, color, and shape for each random distractor. This results in a maximum of 16 objects in each generated world. (For gSCAN, the maximum is 12.)

For size, which is described by relative scalar adjectives [41], we added an additional constraint. If the command contains a size modifier, a world always contains a distractor of a different size (similarly to gSCAN). To avoid vagueness about the intended referents, we ensure that there are only two sizes in that particular world.

As the complexity of distractors increases, there is an increased probability that there could be more than one object in the world that could be the target referent. To ensure a unique solution for all examples, we develop graph-based representations (see Figure 3 for an example) of our shape worlds and use sub-graph matching algorithms to validate examples (see Appendix D for details).

### 4.3 Compositional Splits

ReaSCAN allows us to define a variety of different train/dev/test splits that vary in complexity. Table 3 provides an overview of the splits that we have explored to date. Test splits from category A investigate novel attribute compositions at the command and object level (see Section 6.2), which are adapted from gSCAN. Test splits in category B investigate how a model generalizes to previously unseen co-occurrences of concepts, including both objects and relations (see Section 6.3), unique to

| | Command-World Pairs | | | Exact Match% (Std.) | | | | | |
| | | | | Random | | M-LSTM | | GCN-LSTM | |
| | Train | Dev | Test | Dev | Test | Dev | Test | Dev | Test |
|---|---|---|---|---|---|---|---|---|---|
| gSCAN | 367,933 | 19,282 | 3,716 | - | - | - | 97.69 (0.22) | - | 98.60 (0.95) |
| Simple | 113,967 | 6,318 | 1,215 | 0.17 (0.06) | 0.11 (0.13) | 93.39 (1.97) | 93.64 (2.52) | 98.06 (0.98) | 97.86 (1.27) |
| 1-relative-clause | 340,985 | 18,903 | 3,635 | 0.14 (0.04) | 0.12 (0.02) | 60.68 (3.04) | 61.28 (1.81) | 97.25 (0.68) | 97.19 (0.79) |
| 2-relative-clauses | 549,634 | 30,470 | 5,859 | 0.12 (0.01) | 0.13 (0.03) | 53.08 (13.9) | 52.77 (14.6) | 96.80 (0.82) | 96.85 (0.75) |
| 2-relative-clauses (RD) | 569,835 | 31,590 | 6,075 | 0.16 (0.02) | 0.12 (0.05) | 89.56 (0.66) | 89.81 (0.60) | 98.14 (0.45) | 97.97 (0.48) |
| All | 539,722 | 29,920 | 5,753 | 0.13 (0.02) | 0.14 (0.03) | 78.48 (1.38) | 79.04 (1.24) | 98.78 (0.55) | 98.96 (0.59) |

**Table 2:** ReaSCAN statistics with random splits and performance results of baseline models trained separately for each command pattern. `All` excludes compositional splits. Results are aggregated from 3 independent runs with different random seeds. Performance for gSCAN is from the original papers for `M-LSTM` [14] and `GCN-LSTM` [39].

ReaSCAN. Finally, category C investigates if a model can extrapolate from simple to more complex embedded phrase structures (see Section 6.4).[4]

# 5 Models

We report ReaSCAN experiments with three models. We give high-level descriptions here, and Appendix E provides additional details.

**Random Baseline**    A sequence-generation model that randomly samples actions from our vocabulary and generates action sequences with the same lengths as the actual action sequences. This serves as the lower bound of model performance.

**M-LSTM**    A multimodal LSTM model, which we adapted from a model proposed for gSCAN [14]. This is a sequence-to-sequence (seq2seq) model [42] that takes an encoding of the visual input as a separate modality. The encoder consists of two parts: a bidirectional LSTM (BiLSTM; [43, 44]) as the language encoder for the commands, and a convolutional network (CNN) [45] as the shape-world encoder. Given a world-command pair $(\mathcal{W}_{i,j}, \mathbf{C}_i)$ as the input, the goal is to generate an action sequence $\mathbf{a}_{i,j}$. The output sequence is generated by an attention-based bidirectional LSTM.

**GCN-LSTM**    A graph convolutional neural (GCN) network with a multimodal LSTM which is, to the best of our knowledge, the currently best-performing model on gSCAN [39]. The model encodes commands using a BiLSTM with multi-step textual attention [46]. The shape world is encoded using a GCN layer. The command embedding is fed into the GCN, which makes it language-conditioned. The nodes in the GCN are initialized with representations of the objects in the shape world, where these representations are binary encodings of the objects' attributes. Then, it performs multi-rounds message passing to contextualize object embeddings based on relations. Then, the object embeddings are fed through another CNN layer before feeding into an attention-based BiLSTM together with the command embedding to generate the output sequence, as in Ruis et al. [14].

# 6 Experiments

## 6.1 Random Split

We generate large random splits for all patterns to validate that models can learn to follow ReaSCAN commands when there are no systematic differences between training and testing. We do this while systematically varying the complexity of the inputs, from `Simple` (no relative clauses, as in gSCAN) to `2-relative-clauses`, and we evaluate when merging all three together (`All`). Appendix A provides additional details concerning how these splits are created.

The results in Table 2 show that the `GCN-LSTM` is uniformly superior to the `M-LSTM`. In addition, for both models, performance drops as the number of relative clauses grows. The `M-LSTM` performs far

---

[4]We don't report on some splits from gSCAN, such as novel relative agent positions, novel action length, and novel adverbs, since ReaSCAN introduces only minimal changes for these conditions. However, our ReaSCAN pipeline generates these splits as well.

| Compositional Splits | Command-World Pairs | Exact Match% (Std.) | | |
|---|---|---|---|---|
| | | Random | M-LSTM | GCN-LSTM |
| Simple (Test) | 921 | 0.07 (0.06) | 96.27 (0.54) | 99.71 (0.22) |
| 1-relative-clause (Test) | 2,120 | 0.08 (0.07) | 79.09 (2.63) | 99.14 (0.23) |
| 2-relative-clauses (Test) | 2,712 | 0.10 (0.02) | 73.16 (1.85) | 98.58 (0.54) |
| All (Test) | 5,753 | 0.14 (0.03) | 79.04 (1.24) | 98.96 (0.59) |
| A1:novel color modifier | 22,057 | 0.12 (0.05) | 50.36 (4.03) | 92.25 (0.77) |
| A2:novel color attribute | 81,349 | 0.14 (0.01) | 14.65 (0.55) | 42.05 (4.55) |
| A3:novel size modifier | 35,675 | 0.14 (0.03) | 50.98 (3.69) | 87.46 (2.22) |
| B1:novel co-occurrence of objects | 10,002 | 0.12 (0.03) | 52.17 (1.63) | 69.74 (0.30) |
| B2:novel co-occurrence of relations | 6,660 | 0.16 (0.05) | 39.41 (1.53) | 52.80 (2.75) |
| C1:novel conjunctive clause length | 8,375 | 0.10 (0.01) | 49.68 (2.73) | 57.01 (7.99) |
| C2:novel relative clauses | 8,003 | 0.09 (0.02) | 25.74 (1.36) | 22.07 (2.66) |

**Table 3:** ReaSCAN statistics with compositional splits and performance results of baseline models trained with all command patterns. Results are aggregated from 3 independent runs with different random seeds.

worse with longer clauses (43.65% drop from Simple to 2-relative-clauses). The GCN-LSTM experiences smaller drops (1.03% from Simple to 2-relative-clauses). These results suggest that graph-based neural networks may be better at capturing relations between objects and reasoning over relations than the plain CNNs used by the M-LSTM. Additionally, the GCN-LSTM shows smaller standard deviations from random initializations, suggesting it is more robust on the ReaSCAN task as well.

When we resample shape worlds with only random distractors, the performance of both models increases. In fact, with random distractors, test performance of 2-relative-clauses drops less than 4% compared to the Simple conditions, for both models. This finding reinforces the importance of sampling challenging distractors.

## 6.2 A: Novel Object Attributes

Evaluating neural models on unseen combinations of object attributes remains an ongoing challenge [10, 11, 47]. Here, we extend gSCAN's efforts in this area by testing models on unseen composites of size, color, and shape.

**A1: Novel Color Modifier**   In this split, we hold out all examples where the commands contain "yellow square" for any size (e.g., "small yellow square" or "big yellow square"), meaning that models cannot ground any targets to the expression containing "yellow square". However, the train set includes examples with phrases such as "yellow cylinder" (52,820 unique examples) and "blue square" (90,693 unique examples). At test time, models need to zero-shot generalize in order to interpret "yellow square" correctly. Our distractor sampling strategy ensures that the scenario contains relevant non-yellow squares and non-square yellow things, so that both shape and color information needs to be integrated for correct target identification. Table 3 shows that both models perform worse on these splits than with random splits, with the M-LSTM showing the largest drop in performance. While the GCN-LSTM is clearly getting traction on this task, the results show that compositional generalization remains a serious challenge.

**A2: Novel Color Attribute**   In this split, we test model performance on a novel combination of the target referent's visual features. To test that, we ensure that red squares are never targets during training. Commands also never contain "red square" even in the position of the relations (i.e., inside the relative clauses). However, differently sized red squares are seen during training since they often appear as non-target background objects (266,164 unique examples). We make sure the color attribute is necessary for identifying the target referent, and restrictions apply to objects at all positions in the command. Our results in Table 3 show that this split is slightly harder for both models (with a 81.47% drop for M-LSTM and a 57.51% drop for GCN-LSTM) than A1 as models need to learn visual composites of "red square" from potential reasoning over background objects. Once again, our results suggest that GCN-LSTM is better at generalizing to unseen compositions.

**A3: Novel Size Modifier**   Size is a relative concept in our commands; the same object could be a small square in one context and not in another, depending on the sizes of the other squares present. Similar to A1, we evaluate whether models can zero-shot generalize to new size/shape combinations. Specifically, we hold out all commands containing "small cylinder", meaning that models have not seen expressions such as "small cylinder" or "small yellow cylinder" during training. At test time, models need to generalize when a small cylinder in any color is referred to with expressions such as "small cylinder". During training, the models still learn the relative meaning of "small" by seeing examples containing expressions such as "small square" (22,866 unique examples) or "small red circle" (23,838 unique examples). In addition to generalizing over new composites, models also cannot simply memorize "small" as a specific size (e.g., object of size 2), since the meaning is contextually determined. Similar to A1, we ensure that the size attribute is necessary for identifying the referent target, and restrictions apply to objects at all positions. Table 3 shows that both models achieve comparable performance to A1, which suggests that the generalization capabilities across unseen color and size composites for both models are similar. `GCN-LSTM` continues to perform better than `M-LSTM`, suggesting that it is more successful in generalizing to relative modifiers as well.

## 6.3   B: Novel Co-occurrence of Concepts

In this experimental condition, we assess the ability of models to generalize to novel combinations of concepts, including objects and relations at the clause level.

**B1: Novel Co-occurrence of Objects**   To construct this split, we first collect all objects (e.g., "small red circle" and "big blue square") mentioned in the training set. Then, we construct commands with seen objects that never co-occur during training. Additionally, we control commands to only contain co-occurrences of relations that are seen during training. In this condition, the `GCN-LSTM` continues to outperform the `M-LSTM` in generalizing to unseen co-occurrences of relations. Compared to novel attribute modifiers (i.e., A1 and A3), `GCN-LSTM` performance decreases.

**B2: Novel Co-occurrence of Relations**   In this split, we hold out examples containing commands mentioning both "same size as" and "inside of" relations, meaning the models never see examples such as "walk to the object that is the same shape as the red object and that is inside of the red box". However, in training, models see cases where the relation "inside of" co-occurs with other relations, such as "same row as" (58,863 unique examples). Table 3 shows that both models perform worse compared to B1. This suggests that generalizing over co-occurrence of relations, which requires novel reasoning about objects, is harder for both model architectures.

## 6.4   C: Novel Phrase Structures

As shown in Section 4.1, the number of phrase structures in ReaSCAN can be manipulated. In the following experiments, we test whether a model trained with at most two relative clauses (see Section 4.1 for all patterns) can generalize well to commands with novel phrase structures.

**C1: Novel Conjunctive Clause Length**   In the first experiment, we generate examples with commands that have one additional conjunction clause (i.e., "and $REL_CLAUSE" is added to the `2-relative-clauses` commands). Our results in Table 3 suggest that both models struggle to generalize over longer relative clauses (with a 37.15% drop for `M-LSTM` and a 42.39% drop for `GCN-LSTM`). Since both models are LSTM-based, it may suggest that LSTM-based models don't generalize well to longer sequences at test time, which has been found in more recent studies [12], though some of this may trace to how stop tokens are used [48].

**C2: Novel Relative Clauses**   In this experiment, we generate examples with commands that have two recursive relative clauses (i.e., "and" is swapped with "that is" in the `2-relative-clauses` commands)[5]. For this condition, both models result in catastrophic failures (with a 67.43% drop for `M-LSTM` and a 77.70% drop for `GCN-LSTM`). Our results suggest that GCN is incapable of generalizing over novel recursive relations. The performance degradation of `GCN-LSTM` may suggest that the fault lies with the way the GCN component embeds relational information in its object representations. This is a strength for known combinations but a potential hindrance for novel ones.

---

[5]We only allow relations to be "same row as" and "same column as" to avoid invalid commands.

# 7 Conclusion

We introduced the ReaSCAN benchmark, which seeks to build on the insights behind the gSCAN dataset of Ruis et al. [14] while addressing its shortcomings. ReaSCAN is designed to support controlled assessments of whether models have truly learned grounded, compositional semantics. We find that a state-of-the-art `GCN-LSTM` model achieves strong results for most of the compositional splits from gSCAN. Results on ReaSCAN, however, suggest that those capabilities are overestimates. Furthermore, ReaSCAN allows for more intricate investigations of the resolution of linguistic structure. The `GCN-LSTM` model is successful at tasks involving novel linguistic modifiers and novel entity attribute combinations, but it fails to generalize in settings involving novel relation combinations and longer commands. These results indicate that, while we are making progress in achieving grounded, compositional models, many substantial challenges remain. While ReaSCAN introduces complexity to the problem, via sophisticated distractor sampling strategies and more elaborate input commands, the controlled nature of its input commands means that it is far from tackling the full complexity of natural language. Extending ReaSCAN with interpreted naturalistic English commands would begin to address this limitation.

## Broader Impact

The ReaSCAN benchmark is designed to facilitate the development of models that can use language in a grounded, compositional fashion. Such research has implications for technology development as well as fundamental research in cognitive science and linguistics. We do not foresee any negative impact on society or on the scientific community stemming directly from this research.

## Acknowledgements

This research is supported in part by the National Research Foundation, Singapore, under its AI Singapore Program (AISG Award No: AISG2-RP-2020-016), and in part by a Stanford HAI Hoffman–Yee grant.

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
