# OpenReview forum: "ReaSCAN: Compositional Reasoning in Language Grounding"
_NeurIPS.cc/2021/Track/Datasets_and_Benchmarks/Round1 — NeurIPS 2021 Datasets and Benchmarks Track (Round 1)_

### Official Review · Reviewer_CcLb · 2021-06-29
**Improvements not very clear with results**

**Rating:** 6
**Confidence:** 3
**Clarity:** Yes, I think the paper is clear.

**Strengths:**

The paper proposes complex command generation process which makes ReaSCAN a better alternative to test models on. They also provide better distractor sampling techniques (unlike random sampling for gSCAN) which could help in better analyzing the grounding issues of NLP models.

**Weaknesses:**

It wasn't very clear how ReaSCAN was better than gSCAN quantitatively. I think it would have been better to do some interpretability analysis on the models under experimentation to see what extra information it learnt with the new benchmark. One of the ways to analyze this could be to train on gSCAN and test on ReaSCAN and look for specific model decision making flaws which were mitigated with the new benchmark.

**Additional Feedback:**

Its a nice work but it would make the paper more stronger with more analysis on the value it adds over gSCAN. Theoretically, the improvements make sense but would become more convincing if more metrics and models are brought in the discussion. Its hard for the community to move from one benchmark to other unless the improvements in the new benchmark is significantly better which can be provided with more analysis.

**Correctness:**

The claims made in the paper makes sense qualitatively but require more empirical study to quantify improvements.

**Documentation:**

Based on the code repo link provided in the paper, it seems alright with respect to data collection, availability, and ethical use.

**Ethics:**

I can't think of any ethical issues since the data was synthetically generated.

**Relation To Prior Work:**

Yes, the paper discusses the difference against prior work clearly.

**Summary And Contributions:**

The authors have proposed an improved version of gSCAN to test compositionality and grounding of NLP system, named as ReaSCAN. This new benchmark tries to solve some flaws of gSCAN, such as the irrelevance of word order, biased distractor sampling, etc, hence, claiming it to be more challenging benchmark to test model against. In the experiments, they assess two models, i.e, M-LSTM and GCN-LSTM.

---

> ### Author Response · Authors · 2021-07-10
> **Response to suggestions from Reviewer CcLb**
>
> We thank R4 for the helpful review. We address their comments below along with relevant changes to the paper.
>
> > **Re: Clarifying the ways in which ReaSCAN improves on gSCAN quantitatively.**
>
> The extensive modifications we add into the data generation pipeline of gSCAN make it hard to compare them quantitatively using model performance. One aspect we did not include in our initial draft was the SOTA model (i.e., GCN-LSTM) performance on similar splits across two datasets. For example, [1] proposes the GCN-LSTM model which largely solves unseen testing splits in gSCAN but fails to do so with unseen testing splits of ReaSCAN. For example, in the same novel color modifier/attribute cases (see Sec. 6.2 in the paper), GCN-LSTM achieves 99.08 (unseen “yellow square”) and 80.31 (unseen “red square”) for gSCAN but only achieves 92.25 (unseen “yellow square”) and 42.05 (unseen “red square”) for  ReaSCAN. ReaSCAN reveals that compositional generalizations of novel color modifiers/attributes are far from being solved. We updated the paper to reflect these insights accordingly and added even harder compositional splits that are unique to our ReaSCAN data generation framework (see Sec. 6.3 and Sec. 6.4 in the paper).
> Another important improvement ReaSCAN brings is adding robustness to our unseen testing splits. gSCAN [2] reports extremely high standard deviation for most of its testing splits which makes it hard to interpret the results. We hypothesize that this may reveal artifacts from the data generation process of gSCAN rather than defects of neural models (see Sec. 3 in the paper). Quantitatively as shown in our Table 2 and Table 3, all of our splits result in a much smaller standard deviation even with the same model architectures. It suggests that ReaSCAN may be a more robust and internally coherent dataset for evaluating the model’s generalization skills over linguistic composition.
>
> > **Re: Training a model on gSCAN and testing on ReaSCAN.**
>
> It is technically hard to train with gSCAN, and zero-shot evaluate the model with ReaSCAN. First, the vocabulary and compositions of commands are different across the two datasets. For example, gSCAN does not contain the vocabulary “that”, “is” and “and”. Additionally, gSCAN only contains non-relative-clause cases which makes it unfair to evaluate with commands containing clauses.
>
> **References**
>
> [1] Tong Gao, Qi Huang, Raymond J. Mooney. 2020. “Systematic Generalization on gSCAN with Language Conditioned Embedding” in AACL-IJCNLP 2020.
>
> [2] Laura Ruis, Jacob Andreas, Marco Baroni, Diane Bouchacourt, Brenden M. Lake. 2020. “A Benchmark for Systematic Generalization in Grounded Language Understanding “ in NeurIPS 2020.

---

### Official Review · Reviewer_MrMT · 2021-07-03
**Solid improvement over gSCAN. Lacks evaluation of pretrained NLP models.**

**Rating:** 5
**Confidence:** 4
**Correctness:** Yes, the evaluation method and experi…
**Clarity:** Yes, the paper is well written.

**Strengths:**

1. ReaSCAN is a solid improvement upon gSCAN. The author proposed: 1) a more sophisticated command generator that contains relation clause. Compared with the commands in gSCAN which can be simply modeled via bag-of-words, commands in gSCAN have more complicated linguistic structure and are more challenging to resolve, 2) new algorithms for generating the distractors. There are three types of distractors in ReaSCAN: 1) attribute-based distractors, 2) isomorphism-based distractors, and 3) relation-based distractors. These generated distractors offer a way to control the complexity of queries in ReaSCAN.

2. The author shows that the state-of-the-art GCN-LSTM model cannot generalize in settings that involve more complicated reasoning, e.g., novel color attribute, novel relation co-occurrence, and novel clause length.


**Weaknesses:**

1. ReaSCAN is created on top of gSCAN. Since the goal of gSCAN is also to evaluate compositional generalization, ReaSCAN is a incremental but solid improvement over the prior work.

2. The author has not evaluated the performance of pretrained NLP models like BERT on ReaSCAN. The challenges described in ReaSCAN (e.g., generalization to new color attributes) can also be solved by pretraining the model on a large and diverse corpus. In fact, the success of BERT and GPT-3 shows that pretraining on a large corpus can greatly improve the generalization ability.


**Additional Feedback:**

The benchmark will be much stronger if the author can evaluate the performance of pretrained NLP models on ReaSCAN.

-------- Post Rebuttal -----
I've read the authors' rebuttal and choose to keep my score. My major concern is that the authors have not compared with methods based on pretrained language models, which are currently the SOTA for solving different types of NLP problems. In the real-world scenario, it is easy to obtain a large-scale corpus that contains sentences with diverse linguistic structures. Compared to sentences in real-world applications that may contain grammar errors, spelling mistakes, abbreviations, etc., the synthetic sentences generated in ReaSCAN have clear and interpretable structures and are not very realistic. The methods that are benchmarked are far from real-world solutions for NLP problems that heavily use pretrained models such as word embeddings and contextual embeddings.

**Documentation:**

Yes.

**Ethics:**

No.

**Relation To Prior Work:**

Yes.

**Summary And Contributions:**

ReaSCAN is a synthetic dataset for evaluating the models' ability of compositional language interpretation and reasoning. Each sample in ReaSCAN contains a natural language command, a simulated shape world, and the action sequence that move the object in the shape world following the command. It is an improved and more challenging version of gSCAN. Different from gSCAN, ReaSCAN ensures 1) word-order permutation in commands will affect the final action sequence, 2) a number of distractors are generated so that the model must resolve the syntatic structure of the command in order to generate the correct prediction. The author evaluated two models on ReaSCAN: M-LSTM and GCN-LSTM. Results show that the state-of-the-art GCN-LSTM model fails to generalize in settings that involve novel relation combinations and longer commands.

---

> ### Author Response · Authors · 2021-07-10
> **Response to suggestions from Reviewer MrMT**
>
> We thank R3 for the insightful review. It is helping us to more clearly convey how ReaSCAN marks a notable advance from gSCAN.
>
> > **Re: Quantifying ReaSCAN's improvements over prior work.**
>
> It’s true that ReaSCAN shares the same grid-world setup with gSCAN with synthetic commands. However, we would like to argue that it addresses some of the important limitations of previous works that are worthy for the community to consider when developing new neural models (see Sec. 3 in the current paper version but we’re currently reviewing how we can emphasize these points more explicitly). Although the setup shares commonalities with gSCAN, the underlying process of generating distractors has been completely redesigned. ReaSCAN adds the dimension of syntactic compositionality through a distinct command generation process. In terms of quantitative improvements compared with gSCAN, [1] proposes the GCN-LSTM model which largely solves unseen testing splits in gSCAN but fails to do so with unseen testing splits of ReaSCAN. For example, in the same novel color modifier/attribute cases (See Sec. 6.2 in the paper), GCN-LSTM achieves 99.08 (unseen “yellow square”) and 80.31 (unseen “red square”) for gSCAN but only achieves 92.25 (unseen “yellow square”) and 42.05 (unseen “red square”) for ReaSCAN.
>
> Additionally, we would like to argue that the data generation framework proposed along with ReaSCAN is capable of generating extensive testing splits that are useful in studying linguistic compositions of neural models, which is lacking in gSCAN. For example, unseen cooccurrence in phrase level (see Sec. 6.3 in the paper) and novel relational clauses for both novel length and composition (see Sec. 6.4 in the paper).
>
> >  **Re: Evaluating with pretrained language models.**
>
> We agree that pretrained models generally increase generalizability and that they also provide an interesting case to investigate when it comes to investigating capabilities to interpret novel compositions. To discuss the challenges with evaluating them on ReaSCAN, we first would like to distinguish between what’s commonly referred to as generalizability of pretrained models and the generalizability we are testing with ReaSCAN. For pretrained models, generalizability is usually assessed across domains and tasks. For example, a pretrained model trained with the masked language modeling task with wikipedia data may generalize well with a unseen sentiment analysis task with unseen yelp review dataset. Datasets like ReaSCAN or previous works such as SCAN [2] and gSCAN [3] are testing generalizability over linguistic compositions with systematicity. The task and the data source can be the same during training and evaluation, but composites must be different. More importantly, coming back to the point of evaluating with pretrained models, it makes such evaluation extremely hard without breaking the systematicity, as our unseen composites in the testing splits such as “red square” can appear in the pretraining data. One naive way to mitigate this is to hold out those invalid examples and pretraining our own language models. However, the limited control over the pretraining data can introduce exactly the artifacts that we try to avoid by creating synthetic highly-controlled datasets. For example, a pretraining dataset may already contain sentences with multiple clauses. In sum, we hope that ReaSCAN can contribute to evaluating all kinds of language generation models but the limited control over the pretraining dataset potentially invalidates the held-out test splits which is a challenge we hope to see addressed in future work.
>
>
> **References**
>
> [1] Tong Gao, Qi Huang, Raymond J. Mooney. 2020. “Systematic Generalization on gSCAN with Language Conditioned Embedding” in AACL-IJCNLP 2020.
>
> [2] Brenden Lake and Marco Baroni.  Generalization without systematicity:  On the compositional skills of sequence-to-sequence recurrent networks. ICML 2018
>
> [3] Laura Ruis, Jacob Andreas, Marco Baroni, Diane Bouchacourt, Brenden M. Lake. 2020. “A Benchmark for Systematic Generalization in Grounded Language Understanding “ in NeurIPS 2020.

---

### Official Review · Reviewer_UeTK · 2021-07-04
**Expertly crafted benchmark dataset for compositional language interpretation and reasoning about entities and relations**

**Rating:** 8
**Confidence:** 4

**Strengths:**

The authors present a detailed analysis of the shortcomings of their dataset’s predecessor gSCAN and present a sound approach to addressing them. The resulting new version of the dataset allows for testing on various aspects of compositional language interpretation by presenting a number of different command complexity settings as well as train/test splits to control the type of novel attribute or modulation. Testing a state-of-the-art multi-modal CGN on the dataset reveals its capabilities as a benchmark for compositional language interpretation by demonstrating significant performance drops with increasing command complexity and specific generalisation tests.

The paper is very well written and clearly accounts for the reasoning behind the creation of the presented dataset as well as the details of the algorithms applied in creating it. The tested models are described well and parameter settings are detailed in the supplementary material.

The dataset is freely available on a well-known code-sharing platform and also can easily be re-generated based on the instructions and code in the submission.


**Weaknesses:**

The dataset is created for testing compositional language interpretation in highly abstracted grid worlds. Ideally we would like to get to a point where models can be tested on more complex real-world scenarios - but the significant model shortcomings demonstrated in the paper indicate that we’re simply not there yet.


**Additional Feedback:**

A great contribution to the field!

**Clarity:**

The paper is very well written and clearly presents the dataset and its application as a benchmark for compositional language interpretation.


**Correctness:**

The paper clearly accounts for the reasoning behind the creation of the presented dataset as well as the details of the algorithms applied in creating it. The evaluation of the tested models is sound.


**Documentation:**

The paper accounts for the reasoning behind the creation of the presented dataset as well as the details of the algorithms applied in creating it. The dataset is freely available on a well-known code-sharing platform and also can easily be re-generated based on the instructions and code in the submission.


**Ethics:**

As the dataset is procedurally created using abstract grid-worlds, there are no ethical implications.


**Relation To Prior Work:**

The authors present a very detailed analysis of the shortcomings of their dataset’s predecessor gSCAN but mention other datasets and approaches to linguistic compositionality only in passing.


**Summary And Contributions:**

The paper presents ReaSCAN, a novel benchmark dataset for compositional language interpretation and reasoning about entities and relations that was developed after a thorough analysis of the shortcomings of previously presented datasets. The dataset contains instructions for actions in a grid word that are generated through regular expressions, and the corresponding maps generated under a number of carefully designed distractor sampling regimes.

With their dataset, the authors not only address previous shortcomings of instruction complexity and distractor placement, but also create an opportunity for investigating the effects of different train/test splits that can be used to systematically test model performance on a specific aspect of generalisation difficulty presented by the data.

Comparing a vanilla multi-modal LSTM model with a state-of-the-art graph convolutional neural network, the paper demonstrates the GCN’s superior level of robustness to command complexity and its improved capability to generalise to unseen data aspects, but also clearly reveals its shortcomings with respect to aspects like novel color attributes, novel relation co-occurrence and novel clause length. The dataset therefore appears to be an interesting benchmark for at least these facettes of compositional language interpretation - and could easily be modified to address others, too.

---

> ### Author Response · Authors · 2021-07-10
> **Response to suggestions from Reviewer UeTK**
>
> We thank R2 for the helpful review. We really like the characterization that we are *"helping to chart a path from ReaSCAN to even more ambitious targets"*.
>
> > **Re: Models can be tested on more complex real-world scenarios.**
>
> We absolutely agree. As stated in our response to R1, it’s true that synthetic benchmarks still have limited complexity compared to real-world data. However, our unseen testing splits indicate that current neural models lack generalization abilities even with synthetic datasets. As stated in our responses to R1, we also want to emphasize the advantages of ReaSCAN, where targeted aspects of neural models can be assessed through controlled experiments that are hard to test with real-world datasets.
>
> > **Re: Extension of Related work to present other datasets and approaches to linguistic compositionality in more detail.**
>
> Thank you for this insight which is another valuable extension to R1’s suggestions on the related work section. We are currently working through it, and we only need a bit of time to update the paper to reflect it, since it requires a deep dive into the literature.

---

> > ### Author Response · Authors · 2021-07-13
> > **Updated related works**
> >
> > Related to you concern about the related works section, we have now further updated it. If you think we have missed out any paper or if you have any other suggestion, please let us know and we will be happy to include those papers as well.

---

### Official Review · Reviewer_fDHM · 2021-07-05

**Rating:** 8
**Confidence:** 4

**Strengths:**

- The paper explores an important and exciting topic for compositional reasoning in grounded environments. The task is well motivated and contextualized within progress of prior works.
- The new dataset addresses weaknesses of gSCAN, as is clearly described and illustrated through the paper's both text and figures.
- The paper is very well-written, the synthesis of the dataset is described in detail
- The claims are corroborated by quantitative analysis and comparative evaluation both across models (GCN, multimodal) and between datasets (ReaSCAN and gSCAN). The paper presents a very extensive set of experiments with different splits and conditions.



**Weaknesses:**

For all the SCAN tasks, there is the common limitation of using auto-generated language in simple visual environments which potentially mean that models that perform well in these simplified settings may not scale and demonstrate the same compositional reasoning skills over real-world data (either visual or textual).

However, that weakness is not unique to this particular work, but to the whole line of research, which on the other hand also enjoys from related benefits and advantages, such as controllability over the data, simple process to generate large quantities of it, and a lot of potential supervision signals that could help both for training and evaluation of models for the task.

Having some e.g. natural-language extension of the task (e.g. CLEVR-Human) could be nice and useful.

**Additional Feedback:**

No, see feedback above.

**Clarity:**

The paper is clear and easy to follow, and the writing quality is high. It also includes variety of useful visual samples from the data, plots and visualizations, which all help for the presentation of the new dataset.

**Correctness:**

The claims in the paper are explained in detail and corroborated by quantitative analysis.

**Documentation:**

The dataset has a very detailed github repository with information about the files format, splits, and how to reproduce the data. It also explains how to run the baseline experiments on the dataset and provides the necessary commands. Overall I think the paper does a good job in terms of documentation.

**Ethics:**

I don't identify ethical concerns with regards to this work as it's about following simulated auto-generated in a controlled synthesized environments, with no real data and/or humans responses involved in the collection process.  The paper also does include a broader-impact section and so the author show awareness to these considerations.

**Relation To Prior Work:**

The paper discuss relevant background knowledge about compositionality in general and the SCAN datasets in particular. The paper then does a good job in pointing to the specific three limitations of gSCAN (constrained instructions set, irrelevant distractor objects, and redundant modifiers in commands that aren't necessary for performing the navigation successfully) and explaining how they are addressed in the  new dataset.
In the related work section, the author further discuss in greater detail the different design choices of gSCAN and their impact and implications on the task (e.g.  irrelevance of word order, biased sampling of distractors). This explanations greatly inform the reader of the task and the current limitations needed to be addressed.

However, while most of the related work section focuses on gSCAN, I think that more information and context though about prior neural models that have been explored for the dataset, key modeling findings it has enabled, or potential applications models that work well on gSCAN may have, would be useful to strengthen the related work section.

**Summary And Contributions:**

- The paper proposes a new dataset, called ReaSCAN, that extends gSCAN, but goes further in requiring semantic capabilities of reasoning about entities and relations as well as linguistic skills of compositional interpretation.
- It complements the dataset with experiments to evaluate different models types on the dataset, including a multimodal baselines and a graph convolution network.
- The experiments demonstrate how ReaSCAN is more challenging that the predecessor gSCAN dataset.

Update: After reading the other reviews and the author's response I would like to keep my score!

---

> ### Author Response · Authors · 2021-07-10
> **Response to suggestions from Reviewer fDHM**
>
> We thank R1 for the helpful review!
>
> > **Re: Auto-generated language in simple visual environments may not scale with real-world data.**
>
> We absolutely agree with R1’s assessment. Synthetic benchmarks certainly have a more limited complexity compared to real-world data but, as already stated by R1, synthetic benchmarks such as ReaSCAN also offer advantages, where targeted aspects of neural models can be assessed through controlled experiments. Additionally, our unseen testing splits provide insights into how neural models can be designed to solve realistic NLU tasks: such as generalization capabilities to clause length and compositions.
>
> > **Re: CLEVR-Human version of ReaSCAN as the next step.**
>
> This is a great suggestion, thanks! This is exactly the kind of next step we would like to help work towards. Although the vocabulary and command structures of the presented ReaSCAN dataset are still limited, it’s easily extendable to include more complex expressions. We hope that this enables an easier connection to human annotations that can further inform more complex and realistic commands for novel shape worlds.
>
> > **Re: Extension of the related works section.**
>
> Thank you for this insight and we agree that it would be a useful addition to our paper. We are currently working through it, and we only need a bit of time to update the paper to reflect it, since it requires a deep dive into the literature.

---

> > ### Author Response · Authors · 2021-07-13
> > **Updated related works**
> >
> > As per your suggestion, we have updated the related works section. If you think we have missed out any paper or if you have any other suggestion, please let us know and we will be happy to include those papers as well.

---

> > > ### Comment · Reviewer_fDHM · 2021-07-20
> > > **Thanks!**
> > >
> > > Thank you for updating the paper and for your response!

---

### Author Response · Authors · 2021-07-13
**General response to suggestions from all reviewers**

We thank each reviewer for taking the time to give thoughtful comments. They have led to, what we believe are, significant improvements to the paper, specifically of the Related Works section (as suggested by R1 and R2) and the quantification of ReaSCAN improvements over gSCAN (as suggested by R3 and R4). We were excited to see that our paper was characterized as *“explor[ing] an important and exciting topic for compositional reasoning in grounded environments”* (R1), and as *“present[ing] a detailed analysis of the shortcomings of [our] dataset’s predecessor gSCAN and present[ing] a sound approach to addressing them”* (R2). In our comments below, we elaborate on the specific ways in which we have addressed the concerns raised by the reviewers and the revisions we’ve made to the manuscript.


Both, R1 and R2, commented on the limits of the synthetic nature of our dataset which we recognize and included as a limitation into our manuscript. However, as R1 and R2 point out, our model results also suggest that *“we’re simply not there yet”*  to test using *“more complex real-world scenarios”* (R2). Additionally, we also believe that the control we have over the synthetic nature of the dataset enables insights into models that make it useful for in-depth investigations of models’ compositional generalization capabilities at this stage. However, we agree that it’s important to recognize this shortcoming and we do so more clearly now in the paper.


To address suggestions raised by reviewers, we also made a number of updates to our paper to improve ReaSCAN’s robustness and make it even more challenging. All of our changes are reflected in our publicly available repository with detailed release notes at https://reascan.github.io/. We believe these changes can further help readers to differentiate ReaSCAN from existing datasets and show the value it adds for evaluating models’ compositional generalization capabilities. Here, we list **a few important changes**:

* We added in two new unseen-testing splits leveraging ReaSCAN’s data generation framework. Split B1 enables testing of generalization to novel object co-occurrence in the command which complements B2 which allows for testing of co-occurrence of relational clauses. Split C2 enables testing of generalization to novel relative clauses. Our results show that both presented models struggle to get traction on these tasks, and therefore constitute new challenges for future model development.

* To address R3’s and R4’s concerns about quantifying the improvements of ReaSCAN, we included SOTA model (i.e., GCN-LSTM) performance on similar splits across the two datasets. According to the performance on gSCAN splits, GCN-LSTM seems to have solved the generalization to unseen color modifier compounds (99.08) and has good traction on unseen color attribute compositions (80.31). However, in the more controlled settings in ReaSCAN, GCN-LSTM performance drops to 92.25 for novel color modifier compounds and even to 42.05 with novel color attribute compositions. Using more sophisticated distractor sampling and closer control over the sampled splits, ReaSCAN uncovers compositionality challenges that could not be picked up by gSCAN results. This again supports the value of ReaSCAN for assessing models’ capabilities to generalize.

We also updated our main performance result tables (see Table 2 and 3 in the paper) after fixing an implementation issue that resulted in ambiguity over referent targets in some command-world pairs. The GCN-LSTM model improved for random splits, but both models continue to fail to solve our unseen testing splits.

Other minor revisions: We ensured that the referent target is always preceded by the definite determiner “the”, and fixed small typos throughout our manuscript.

We would like to thank again all the reviewers for engaging with us and our paper, and for their helpful comments and suggestions.

---

### Note · ~Zhengxuan_Wu1 · 2021-10-29

https://reascan.github.io/

---

### Decision · Program_Chairs · 2021-07-26

**Decision:**

Accept

**Comment:**

This paper presents ReaSCAN, which extends gSCAN dataset which aims to test grounded compositional semantics. The paper (1) concretely points to the deficiencies of the previous version of the dataset -- irrelevance of word order, a limited test of compositionality, limited distractors, and (2) designs new grammar to synthetically generate examples that are more complex and addresses these deficiencies.

In addition to improving data, they also introduce a carefully designed train test split to test "learning" compositionality. They present two baselines used in prior work, and their experimental results show that this newer benchmark poses unsolved challenges.

While reviewers pointed out the limitation of this work as a synthetic dataset -- not reflecting the real usages of human language --, many pointed out their careful experimental designs and linguistic complexity it brings will be beneficial for the research community to study grounded compositional semantics.